# A DFT Study of the Copolymerization of Methyl Vinyl Sulfone and Ethylene Catalyzed by Phosphine–Sulfonate and α-Diimine Palladium Complexes

**Ling Zhu** [1], **Shuang Li** [1], **Xiaohui Kang** [2,*], **Wenzhen Zhang** [1] **and Yi Luo** [1,3,*]

[1] State Key Laboratory of Fine Chemicals, School of Chemical Engineering, Dalian University of Technology, Dalian 116024, China; 18769478041@163.com (L.Z.); wyclswz@163.com (S.L.); zhangwz@dlut.edu.cn (W.Z.)
[2] College of Pharmacy, Dalian Medical University, Dalian 116044, China
[3] PetroChina Petrochemical Research Institute, Beijing 102200, China
[*] Correspondence: kangxh@dmu.edu.cn (X.K.); luoyi@dlut.edu.cn (Y.L.)

**Abstract:** Density functional theory (DFT) calculations were comparatively carried out to reveal the origins of different catalytic performances from phosphine–benzene sulfonate (**A**, [{P^O}PdMe(L)] (P^O = $K^2$-P,O-Ar$_2$PC$_6$H$_4$SO$_3$ with Ar = 2-MeOC$_6$H$_4$)) and α-diimine (**B**, [{N^N}PdMe(Cl)] (N^N = (ArN=C(Me)-C(Me)=NAr) with Ar = 2,6-$^i$Pr$_2$C$_6$H$_3$)) palladium complexes toward the copolymerization of ethylene and methyl vinyl sulfone (MVS). Having achieved agreement between theory and experiment, it was found that the favorable 2,1-selective insertion of MVS into phosphine–sulfonate palladium complex **A** was due to there being less structural deformations in the catalyst and monomer. Both the MVS and ethylene insertions were calculated, and the former was found to be more favorable for chain initiation and chain propagation. In the case of α-diimine palladium system **B**, the resulting product of the first MVS insertion was quite stable, and the stronger O-backbiting interaction hampered the insertion of the incoming ethylene molecule. These computational results are expected to provide some hints for the design of transition metal copolymerization catalysts.

**Keywords:** density functional theory; phosphine–benzene sulfonate; α-diimine; palladium; methyl vinyl sulfone; copolymerization

## 1. Introduction

Polyethylene is the most prevalent material used in human life, because it comprises a large majority of the 400 million tons of worldwide plastics produced [1,2]; however, its non-polar nature is one of its biggest disadvantages. The coordination–insertion copolymerization of olefin with polar vinyl monomers is considered to be a promising method to enhance the surface properties of nonpolar polyolefins, such as their adhesiveness, printability, dyeability, compatibility, etc. [3–5]. In these reactions, the choice of catalytic system is of paramount importance because the heteroatoms of polar groups are prone to poison the most organometallic catalysts. Late transition metal complexes have attracted much attention, owing to their low oxophilicity in comparison to early transition metals. For instance, the α-diimine Pd (II) catalysts reported by Brookhart (Scheme 1, **I**) have been proven to be a seminal discovery [6,7]. These types of palladium catalysts have been generally limited to a narrower scope of polar monomer substrates, and polar functional groups are incorporated at the ends of their branches. Subsequently, the Drent-type phosphine–sulfonate palladium catalysts (Scheme 1, **II**) were demonstrated to mediate the copolymerizations of ethylene and a broad scope of polar monomers such as acrylates, acrylonitrile, vinyl acetate, vinyl halides, acrylic acid, and vinyl ethers [8].

**Scheme 1.** Recently developed bidentate neutral and cationic palladium-based catalysts.

In addition, to expand the scope of polar monomers (monomers containing heteroatoms other than O atoms), a series of palladium catalysts bearing different frameworks was synthesized. As an important extension, the iminopyridyl Pd (II) catalysts **III** [9,10] reported by Dai et al. showed significant catalytic activity in the copolymerization of ethylene with 2,2,3,4,4,4-hexafluorobutyl acrylate (6FA), yielding high-molecular-weight functionalized polyethylene. An extensive family of Pd (II) catalysts with bisphosphine monoxide-type ligands was developed for ethylene homo- and copolymerization reactions. After this initial advance, significant efforts by Jian and Nozaki were made to develop palladium catalysts series **IV** [11,12], **V** [13,14], and **VI** [15], which could also copolymerize ethylene with a series of polar vinyl monomers containing silicon, chlorine, and nitrogen. Moreover, palladium complexes bearing a bidentate ligand with NHC-phosphine oxide **VII** [16] and palladium/IzQO complexes **VIII** [17] were reported as being important catalysts for the preparation of functionalized polyolefins.

The aliphatic polysulfones, which possess good chemical and thermal stability, have some applications in hydrocarbon fuels, medical devices, and electron-beam fabrication. Such polymers are readily obtained by the free-radical polymerization of olefin and $SO_2$ monomers, or by the gamma irradiation of gaseous monomers. By contrast, coordination–insertion polymerization as a controlled method would be a more attractive method to prepare the polysulfones. Thus far, only two studies on the Pd-catalyzed copolymerization of ethylene with methyl vinyl sulfone (MVS) have been reported [18,19]. Li et al. reported the studies of ethylene/MVS copolymerization by palladium catalysts based on a phosphine benzene sulfonate ligand (**A**) and an α-diimine ligand (**B**) (Scheme 2) [19]. As reported in the study, the phosphine benzene sulfonate palladium catalyst **A** obtained linear copolymers with polar units located at both the main chain and chain end. Meanwhile, α-diimine palladium catalyst **B** was inactive in above-mentioned reactions. The origin of the different activities of complexes **A** and **B** remained unclear. To enhanced the catalytic activity, numerous modifications of the sterically bulky *N-o*-aryl substituent and ligand backbone have been developed. For example, catalysts bearing α-diimine ligands with camphyl-derived backbones displayed modest activity for ethylene polymerization, as well as at high temperature [20], and a positive effect on CO/styrene copolymerization was observed when the phenyl groups on the imine nitrogen were replaced with bis(imino)acenaphthene [21]. Encouraged by previous experimentation and theoretical works [22–24] on the organometallic complex-catalyzed olefin polymerization, a systematic computational study was conducted for the Pd-catalyzed ethylene/MVS copolymerization to clarify the detailed reaction mechanism and the origin of distinct differences between **A** and **B**. On the basis of the calculated results, a further series of novel catalysts with higher activity were designed.

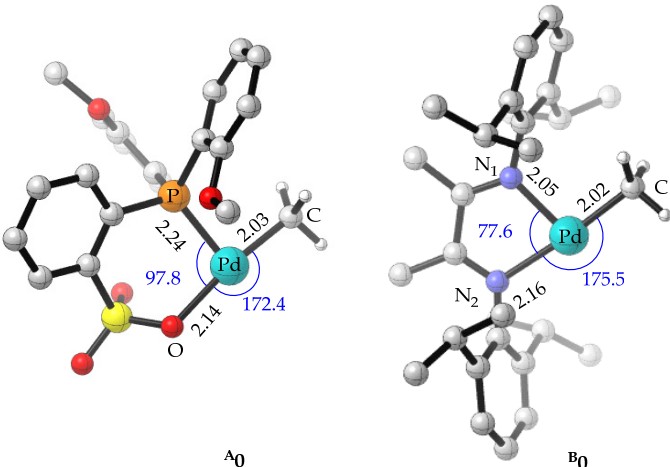

**Scheme 2.** Copolymerization of ethylene/MVS by catalysts **A** and **B**.

## 2. Results and Discussion

### 2.1. Structures of Active Species $^A0$ and $^B0$

Firstly, the active species [{P^O}PdMe] (P^O = $K^2$-P,O-$Ar_2PC_6H_4SO_3$, Ar = 2-$MeOC_6H_4$) $^A0$, and [{N^N}PdMe]$^+$ (N^N = (ArN=C(Me)-C(Me) = NAr), Ar = 2,6-$^iPr_2C_6H_3$) $^B0$ were computationally optimized. As shown in Figure 1, the optimized neutral species $^A0$ with the Pd–Me group cis to the P atom was formed by the labile ligand (L) dissociating from the phosphine–benzene sulfonate palladium catalyst **A**. [25–27] Geometrically, the tri-coordinated T-shaped geometry was retained in species $^A0$, as suggested by the lengths of the Pd–C (2.03 Å), Pd–O (2.14 Å), and Pd–P (2.24 Å) bonds, as well as by the angles of ∠C-Pd-O (172.4°) and ∠P-Pd-O (97.8°). Meanwhile, cationic species $^B0$ showed similar geometrical features, as suggested by the average Pd–$N_{avg}$ ($N_1$ and $N_2$) bond length of 2.11 Å, as well as by the angles of ∠C-Pd-$N_2$ (175.5°) and ∠$N_1$-Pd-$N_2$ (77.6°). The species $^A0$ and $^B0$ indicated nearly the same lengths of the Pd–C bonds (2.03 Å in $^A0$ and 2.02 Å in $^B0$). Based on these two species, $^A0$ and $^B0$, detailed copolymerization mechanisms of ethylene with MVS were computationally considered.

**Figure 1.** Geometric structures (distances (black numbers) in Å and angles (blue numbers) in degree) of active species $^A0$ and $^B0$. The hydrogens, excepting for those on the methyl group, are omitted for clarity.

### 2.2. Copolymerization Mechanism of Ethylene and MVS by the Phosphine–Benzene Sulfonate Catalyst **A**

#### 2.2.1. Chain Initiation

Firstly, the copolymerization mechanism of MVS and ethylene (E) by active species $^A0$ was computed. As shown in Figure 2, an ethylene coordinates with the three-coordinate palladium (II) complex $^A0$ to yield a π-complex $^A1_E$ by releasing the energy of 9.6 kcal/mol.

The orientation of the ethylene in the π-complex $^A1_E$ is such that the double bond is perpendicular to the Pd-P-O square plane and locates trans to the P atom. As previously reported for phosphine–sulfonate-based catalysts [25–27], the *cis/trans* isomerization from $^A1_E$ to the more reactive isomer $^A3_{Eiso}$ with the Me group trans to the P atom is necessary for further ethylene insertion. Starting from $^A1_E$, two possible *cis/trans* isomerization pathways (Figures S1 and 2) were found. One pathway is that the direct isomerization from $^A1_E$ to $^A3_{Eiso}$ goes through a $^ATS0_{Eiso}$ via an energy barrier of 22.2 kcal/mol (see the Figure S1 in the Supplementary Materials (SM)). The other pathway is that the MeO group associates to the Pd center via a barrierless transition state $^ATS1_{Eiso}$ to yield intermediate $^A2_{Eiso}$. Then, the isomerization of the coordinated ethylene and the Me group happens via $^ATS2_{Eiso}$ to generate an intermediate $^A3_{Eiso}$ accompanied by the dissociation of the MeO group from the metal center. This two-step isomerization overcomes an energy barrier of 20.2 kcal/mol. By contrast, the later mechanism is more kinetically favorable than that of a one-step process, confirming that the MeO coordination assists the *cis/trans* isomerization. Furthermore, the rotation of ethylene in $^A3_{Eiso}$ occurs to yield $^A4_E$ with the ethylene parallel into the ligand's Pd-P-O plane, which has a free energy barrier of 6.4 kcal/mol. Then, the ethylene inserts into the Pd–Me bond via $^ATS3_E$ to generate a β-agostic complex $^A6_E$. The insertion of ethylene as the rate-determining step overcomes a total energy barrier of 26.4 $(12.0 - (-14.4))$ kcal/mol.

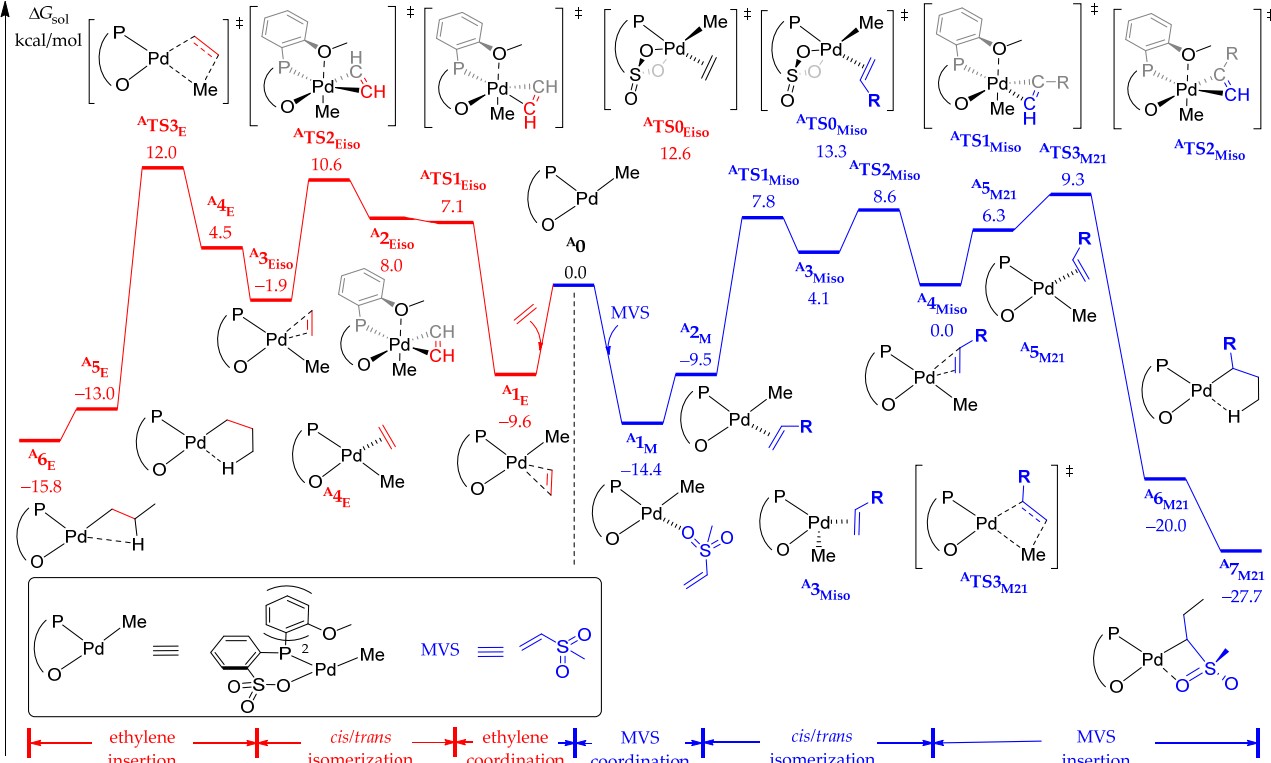

**Figure 2.** Calculated Gibbs free energy profiles of chain initiations of MVS and ethylene mediated by **A** (in kcal/mol).

In the monomer MVS case, the MVS coordinates with the Pd center to form an O-σ coordinating complex $^A1_M$ by the energy release of 14.4 kcal/mol, which is then isomerized to the C=C bond coordinated intermediate $^A2_M$. Based on $^A2_M$, the isomerization of $^A2_M$ to $^A4_{Miso}$ through a two-step pathway ($^A2_M \rightarrow {}^ATS1_{Miso} \rightarrow {}^A3_{Miso} \rightarrow {}^ATS2_{Miso} \rightarrow {}^A4_{Miso}$) happens via an energy barrier of 18.1 $(8.6 - (-9.5))$ kcal/mol, which has been confirmed to be more favorable than that of the one-step process (22.8 $(13.3 - (-9.5))$ kcal/mol, see Figure S2 in the SM). This shows a good agreement with that of the ethylene case where

the coordination of the MeO group with the Pd center facilitates *cis/trans* isomerization. Starting from $^A4_{Miso}$, the regioselectivity in MVS insertion reactions has been extensively studied. As shown in Figure 3, the prochiral MVS insertion with two manners of two enantiofaces were considered, viz. 1,2-manner (1,2-*re* and 1,2-*si*) and 2,1-manner (2,1-*re* and 2,1-*si*). It was found that the transition state (TS) of MVS insertion in (2,1-*si*) mode shows the lowest free energy (9.3 vs. 12.6, 11.9, and 12.7 kcal/mol) in comparison with the other manners, suggesting that 2,1-*si*-insertion of MVS is more kinetically favorable than other modes (the energy of the product of the four insertion manners are shown in Figure S3, in the SM). To shed more light on the superiority of 2,1-*si*, distortion–interaction analyses [28–31] for TSs $^A\text{TS3}_{M21}$ and $^A\text{TS3}_{M12\text{-}si}$ were carried out. As shown in Figure 4, the MVS moiety and the remaining (P^O) PdMe part in these two transition states were called the fragment mono (highlighted in green) and fragment cat., respectively. It is obvious that the total deformation energies were 54.0 (36.0 + 18.0) kcal/mol for $^A\text{TS3}_{M21}$ and 60.8 (39.7 + 21.1) kcal/mol for $^A\text{TS3}_{M12\text{-}si}$, whereas the interaction energies between these two fragments were computed to be −59.0 and −62.8 kcal/mol for $^A\text{TS3}_{M21}$ and $^A\text{TS3}_{M12\text{-}si}$, respectively. These results suggest that the less geometric deformation led to the lower energy of $\Delta E_{TS} = -5.0$ kcal/mol, further stabilizing the $^A\text{TS3}_{M21}$. As shown in Figure 4, further geometric analyses confirmed that the smaller deformation in $^A\text{TS3}_{M21}$ was indicated by the smaller dihedral angle∠Pd-$C_1$-$C_2$-$C_3$ (4.44° in $^A\text{TS3}_{M21}$ vs. 9.73° in $^A\text{TS3}_{M12\text{-}si}$) and the shrinking size of the six-membered -Pd-P-$C_4$-$C_5$-S-O– ring, as suggested by the angles Pd-P-$C_4$ (111.0° in $^A\text{TS3}_{M21}$ vs. 112.7° in $^A\text{TS3}_{M12\text{-}si}$) and the dihedral angles ∠P-Pd-O-S (38.2° in $^A\text{TS3}_{M21}$ vs. 42.7° in $^A\text{TS3}_{M12\text{-}si}$)/∠$C_5$-S-O-Pd (−78.9° in $^A\text{TS3}_{M21}$ vs. −81.2° in $^A\text{TS3}_{M12\text{-}si}$). In addition, there was remarkably smaller root mean square deviation (RMSD) in atomic positions of the fragment monomer and catalyst in $^A\text{TS3}_{M21}$ (Figure S4 (in the SM), catalyst: 0.068 Å and monomer: 0.236 Å) than that in $^A\text{TS3}_{M12\text{-}si}$ (catalyst: 0.128 Å and monomer: 0.269 Å). The above data confirmed that the less repulsive interaction between the catalyst and MVS in the 2,1-insertion manner explains the stability of $^A\text{TS3}_{M21}$ well. Based on this, the 2,1-*si* insertion was selected for the discussion of ethylene and MVS copolymerization. Starting from $^A4_{Miso}$, the 2,1-*si* insertion of MVS had a barrier of 9.3 (9.3 − 0.0) kcal/mol via transition state $^A\text{TS3}_{M21}$. In the whole chain initiation of MVS and ethylene (Figure 2), the 2,1-insertion of MVS showed a lower energy barrier of 23.7 (9.3 − (−14.4)) kcal/mol than that for ethylene insertion (26.4 (12.0 − (−14.4))) kcal/mol). This is consistent with the experimental observation that the polar-initiated chain end segment was observed in the copolymer produced by catalyst **A**.

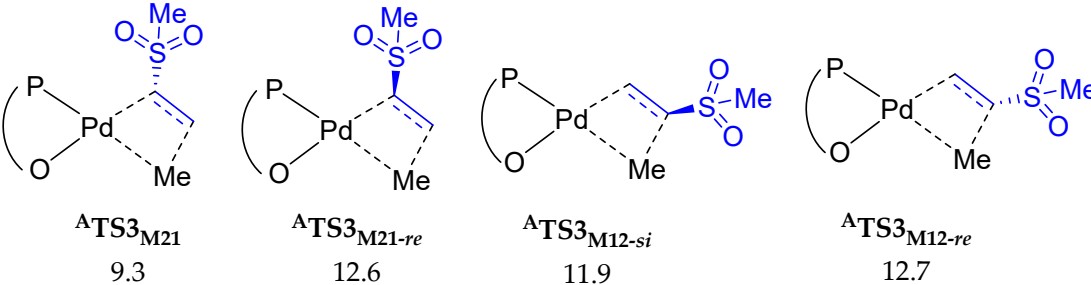

**Figure 3.** Four different enantioselective insertion transition states of MVS into the Pd–Me bond by complex **A** (in kcal/mol).

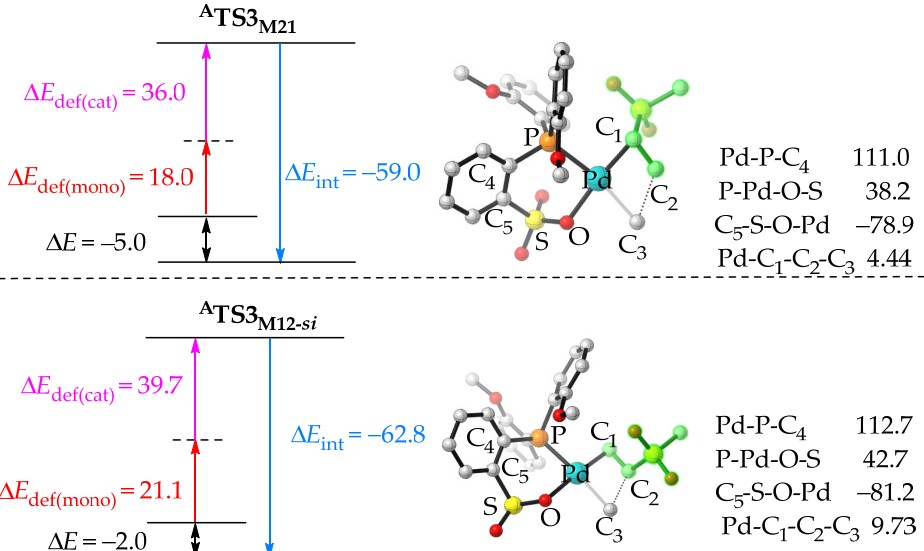

**Figure 4.** Distortion–interaction analyses (kcal/mol) and the geometrical structures (angles in degrees) of $^A$**TS3$_{M21}$** and $^A$**TS3$_{M12-si}$**. All hydrogen atoms are hidden for clarity.

### 2.2.2. Chain Propagation

On the basis of chain initiation products $^A$**6$_E$** and $^A$**7$_{M21}$**, the chain propagation process was also computationally studied. The processes of chain propagation also underwent olefin coordination, *cis/trans* isomerization, and olefin insertion. The computational results are summarized in Figure 5. Starting with $^A$**6$_E$** (Figure 5a), the ethylene coordinated complex $^A$**7$_{EE}$** (−20.6 kcal/mol) was less stable than the MVS-coordinated $^A$**7$_{EM}$** (−25.4 kcal/mol). The sequential isomerization of $^A$**7$_{EE}$** to $^A$**9$_{EE}$** could take place through *cis/trans* isomerization ($^A$**TS4$_{EEiso}$** and $^A$**TS5$_{EEiso}$**) with an energy barrier of 17.3 kcal/mol. Similarly, *cis/trans* isomerization via $^A$**TS4$_{EMiso}$** and $^A$**TS5$_{EMiso}$** required an activation energy of 23.4 kcal/mol relative to $^A$**7$_{EM}$** to afford complex $^A$**10$_{EM}$**. Subsequently, the insertion of ethylene into $^A$**9$_{EE}$** overcame an energy barrier of 14.1 (−0.1 − (−14.2)) kcal/mol, while MVS insertion into $^A$**10$_{EM}$** showed an energy barrier of 9.7 (−4.3 − (−14.0)) kcal/mol. By contrast, the ethylene insertion was the rate-determining step, while for MVS, the *cis/trans* isomerization became the rate-determining step. In the entire energy profiles, the free energy barrier for the ethylene reaction (from $^A$**7$_{EM}$** to $^A$**TS6$_{EE}$**) was 25.3 (−0.1 − (−25.4)) kcal/mol, and was slightly higher than the MVS-involved process (23.4 (−2.0 − (−25.4))) kcal/mol, from $^A$**7$_{EM}$** to $^A$**TS4$_{EMiso}$**), revealing that MVS could incorporate into the polyethylene chain.

On the basis of $^A$**7$_{M21}$**, an ethylene coordination to $^A$**7$_{M21}$** separated the O-backbiting interaction from the metal center to yield $^A$**8$_{ME}$** by absorbing an energy of 1.5 kcal/mol (Figure 5b). Then, the *cis/trans* isomerization from $^A$**8$_{ME}$** to $^A$**10$_{ME}$** took place through a free energy barrier of 20.1 (−6.1 − (−26.2)) kcal/mol. Lastly, ethylene insertion occurred via a 16.4 kcal/mol energy barrier, leading to the product $^A$**12$_{ME}$** with an ethylene–MVS copolymer unit. This whole process had an overall free energy barrier of 21.8 (−5.9 − (−27.7)) kcal/mol, which was lower than that of the MVS-inserting chain initiation (23.7 kcal/mol). The results also suggested that an MVS-E copolymer unit could be achieved.

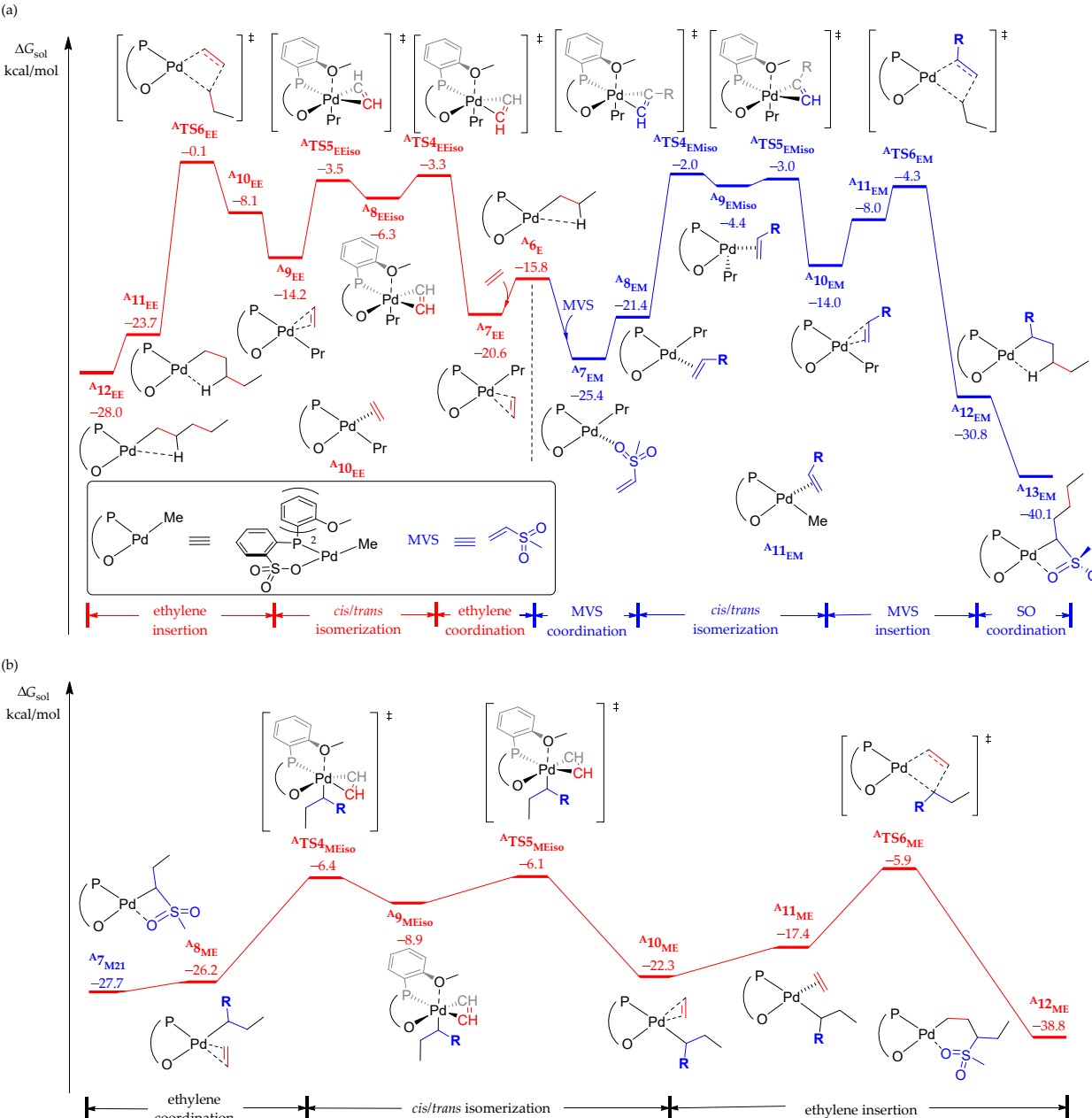

**Figure 5.** Calculated Gibbs free energy profiles of chain propagations based (**a**) $^A6_E$ and (**b**) $^A7_{M21}$ mediated by **A** (in kcal/mol).

## 2.3. Copolymerization Mechanism of Ethylene and MVS by the α-Diimine Palladium Catalyst **B**

In sharp contrast to the phosphine–benzene sulfonate palladium, the typical α-diimine palladium catalyst **B** has been reported to be inactive for the copolymerization of ethylene with MVS [19]. In order to clarify the difference, the chain initiation (Figure 6) and chain propagation (Figure 7) processes in the copolymerization of ethylene with MVS medicated by model species **B** were also systematically calculated.

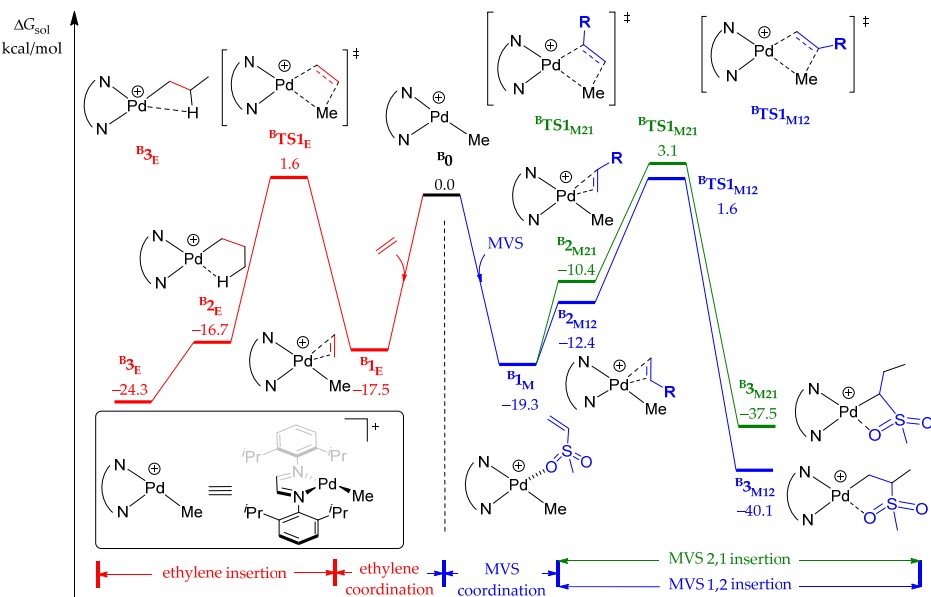

**Figure 6.** Calculated Gibbs free energy profiles of chain initiations of MVS and ethylene mediated by **B** (in kcal/mol).

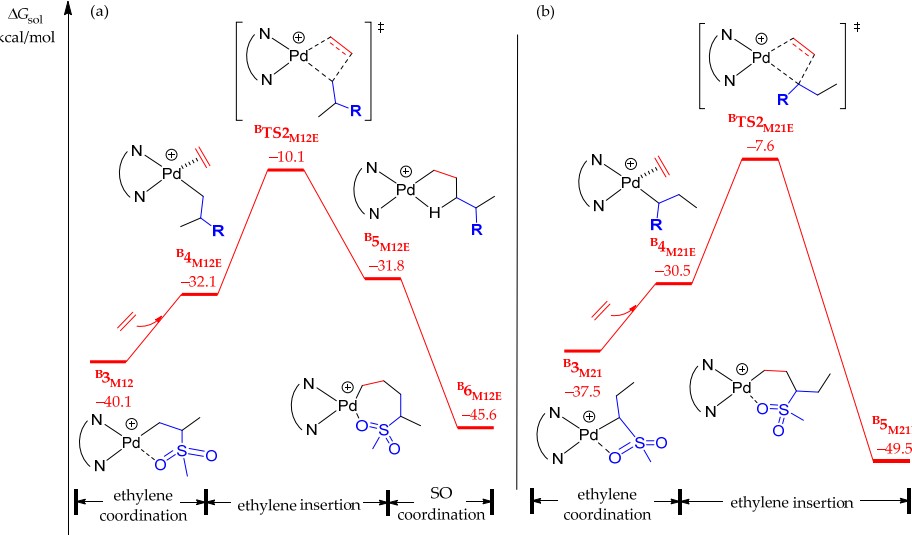

**Figure 7.** Calculated Gibbs free energy profile of chain propagation based on (**a**) $^{B}3_{M12}$ and (**b**) $^{B}3_{M21}$ mediated by **B** (in kcal/mol).

### 2.3.1. Chain Initiation

During the chain initiation process (Figure 6), an ethylene firstly coordinates with species $^{B}0$ to form a π-complex $^{B}1_{E}$, then inserts into the Pd–C bond via $^{B}TS1_{E}$ to yield a γ-agostic propyl complex $^{B}2_{E}$, which finally converts to a more stable β-agostic complex $^{B}3_{E}$. The aforementioned transformation overcomes an energy barrier of 20.9 (1.6 − (−19.3)) kcal/mol, and is easy to occur. Starting from the oxygen-coordinated complex $^{B}1_{M}$, two possible means of insertion of MVS were calculated, viz., the 1,2-($^{B}1_{M} \rightarrow {}^{B}2_{M12} \rightarrow {}^{B}TS1_{M12} \rightarrow {}^{B}3_{M12}$) and 2,1-insertion ($^{B}1_{M} \rightarrow {}^{B}2_{M21} \rightarrow {}^{B}TS1_{M21} \rightarrow {}^{B}3_{M21}$). By contrast, the 1,2-insertion of MVS went through a free energy barrier of 20.9 (1.6 − (−19.3)) kcal/mol, which was slightly lower than that (22.4 (3.1 − (−19.3)) kcal/mol) of the 2,1 case. Therefore, the 1,2-insertion by species **B** is more kinetically favorable in comparison with 2,1-insertion, which is different from the **A**-catalyzed situation. The difference (1.5 kcal/mol) in the insertion energy

barrier between the 1,2-insertion and 2,1-insertion processes was small, suggesting that 2,1-insertion could be also possible.

### 2.3.2. Chain Propagation

Therefore, based on $^B3_{M21}$ and $^B3_{M12}$, the Gibbs energy profiles for the second insertion of ethylene were systematically investigated (Figure 7). The computational results showed that the subsequent ethylene insertions into $^B3_{M12}$ ($^B3_{M12} \rightarrow {}^B4_{M12E} \rightarrow {}^BTS2_{M12E} \rightarrow {}^B5_{M12E}$) and $^B3_{M21}$ ($^B3_{M21} \rightarrow {}^B4_{M21E} \rightarrow {}^BTS2_{M21E} \rightarrow {}^B5_{M21E}$) surmounted the energy barriers of 30.0 ($-10.1 - (-40.1)$) and 29.9 ($-7.6 - (-37.5)$) kcal/mol, respectively, which are both difficult to overcome at room temperature. Therefore, these calculations suggest that ethylene insertion after MVS insertion is impossible, which is consistent with experimental results.

### 2.4. Comparisons of **A**- and **B**-Medicated Copolymerization of Ethylene and MVS

To explore the differences in the catalytic activities of **A** and **B** in ethylene and MVS copolymerization, the chain propagation processes based on products $^A7_{M21}$, $^B3_{M12}$, and $^B3_{M21}$ were compared. It was found that the total energy barriers of chain propagation were 21.8, 30.0, and 29.9 kcal/mol, respectively, which consisted of the coordinated energies of 1.5, 8.0, and 7.0 kcal/mol, respectively, and relative inserted energy barriers of 20.3, 22.0, and 22.9 kcal/mol, respectively (Figure 8). Obviously, the ethylene coordination and ethylene insertion were more difficult in catalyst **B** compared to catalyst **A.** To clarify the origin of the coordination differences, the geometric structures of the chain initiation product ($^A7_{M21}$, $^B3_{M12}$, and $^B3_{M21}$) were analyzed (Figure 9a). The geometrical characters associated with the O-backbiting interactions in $^A7_{M21}$, $^B3_{M12}$, and $^B3_{M21}$ indicated that the $Pd \cdots O_1$ distance was 2.25 Å in $^A7_{M21}$, which was farthest in comparison with those in $^B3_{M12}$ (2.10 Å) and $^B3_{M21}$ (2.10 Å). Therefore, these stronger O-backbiting interactions in $^B3_{M12}$ and $^B3_{M21}$ suppressed the ethylene coordination. In addition, the angles of $\angle N_1$-Pd-$O_1$ in $^B3_{M12}$ (175.8°) and $^B3_{M21}$ (176.2°) were larger than that of $\angle P$-Pd-$O_1$ in $^A7_{M21}$ (167.2°), which also proved the existence of the stronger O-backbiting interactions in $^B3_{M12}$ and $^B3_{M21}$. As we know, charge dispersion is closely connected with the stability of a structure. The unsigned average charges ($|Q|$) and square errors $S$ of Pd, $C_1$, $S_1$, and $O_1$ (or Pd, $C_1$, $C_2$, $S_1$, and $O_1$) atoms in intermediates $^A7_{M21}$, $^B3_{M12}$, and $^B3_{M21}$ were calculated in this study, in order to estimate the degree of charge dispersion. In general, the smaller the value of $S$, the more stable the structure. The results showed that the $S$ value in intermediates $^A7_{M21}$, $^B3_{M12}$, and $^B3_{M21}$ were 0.428, 0.361, and 0.364, respectively. As expected, the $S$ values in $^B3_{M12}$ and $^B3_{M21}$ were smaller than that of $^A7_{M21}$, suggesting that the former are more stable than the latter. This is consistent with the results from the geometric analysis. On the basis of these results, we speculated that increasing the $S$ value of the MVS insertion product perhaps increases the polymerization reactivity.

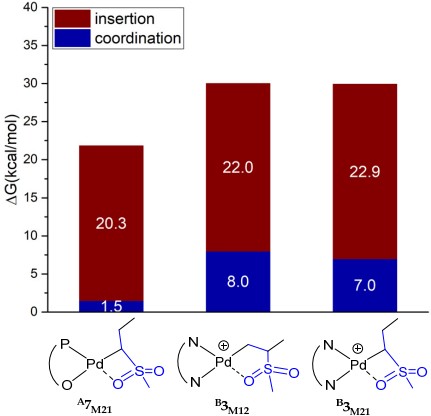

**Figure 8.** Components of the overall chain propagation barriers of the two catalysts.

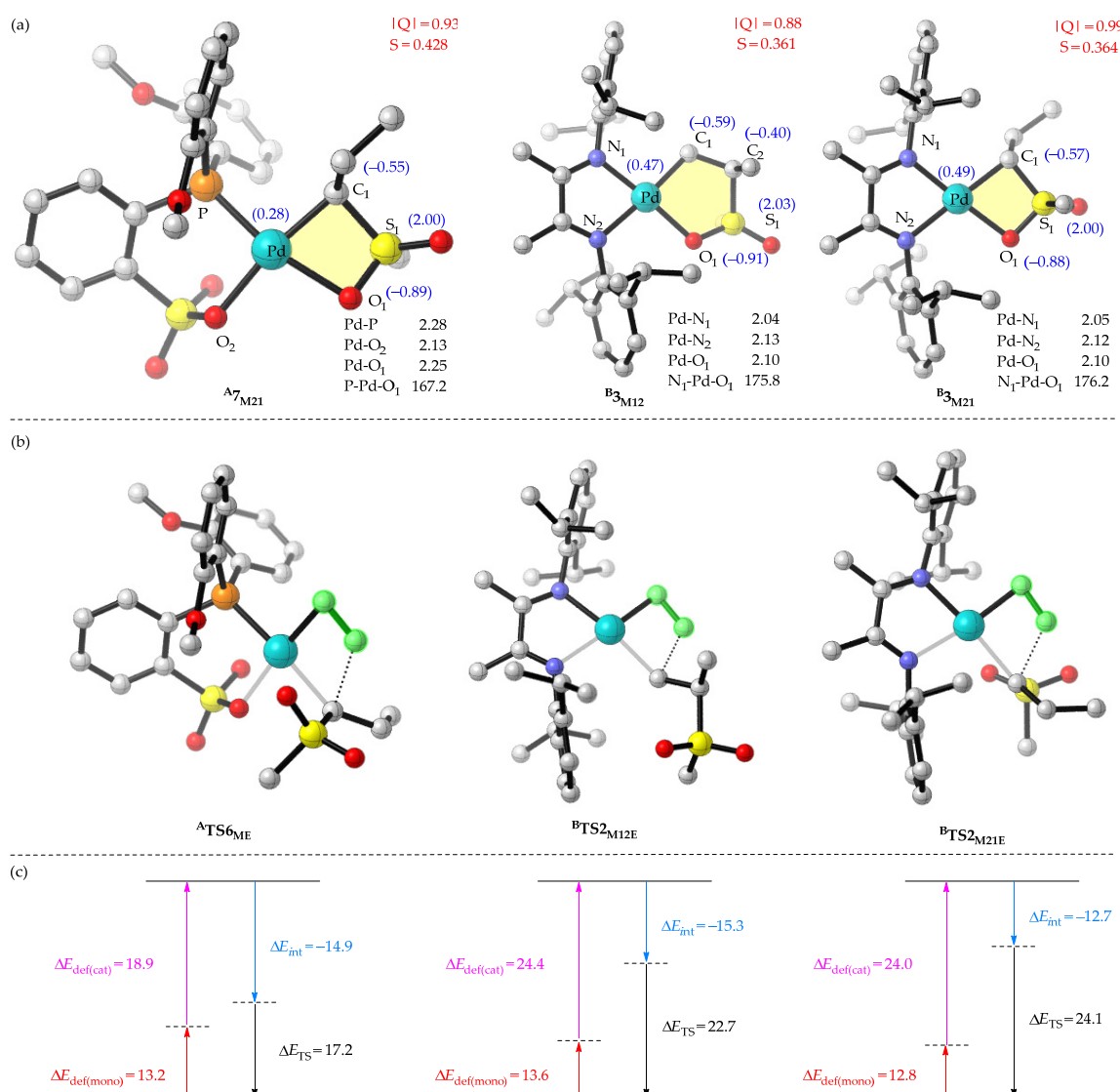

**Figure 9.** (**a**) Geometrical structures (distances (black numbers) in Å, angles (black numbers) in degrees, and NBO atomic charges (blue numbers in parentheses)) of **$^A7_{M21}$**, **$^B3_{M12}$**, and **$^B3_{M21}$**. (**b**) Geometrical structures of transition states **$^ATS6_{ME}$**, **$^BTS2_{M12E}$**, and **$^BTS2_{M21E}$**. (**c**) Distortion–interaction analysis for **$^ATS6_{ME}$**, **$^BTS2_{M12E}$**, and **$^BTS2_{M21E}$** (in kcal/mol). The $S$ value denotes the average square error of charge ($S = \sum (|Q_x| - |Q|)^2/n$, n = 5 or 4), where $|Q|$ represents the unsigned average charge, and $Q_x$ denotes the charge on each atom included. All hydrogens atoms are hidden for clarity.

To further clarify the origin of insertion differences, distortion–interaction analyses [28–31] for TSs **$^ATS6_{ME}$**, **$^BTS2_{M12E}$**, and **$^BTS2_{M21E}$** were carried out (Figure 9c). The analysis scheme used was similar to that for **$^ATS3_{M21}$** and **$^ATS3_{M12-si}$** (*vide ante*). It was found that the total deformation energies $\Delta E_{def}$ in **$^ATS6_{ME}$**, **$^BTS2_{M12E}$**, and **$^BTS2_{M21E}$** were 32.1 (18.9 + 13.2), 38.0 (24.4 + 13.6), and 36.8 (24.0 + 12.8) kcal/mol, respectively, whereas the interaction energies between these two fragments were computed to be −14.9, −15.3, and −12.7 kcal/mol for **$^ATS6_{ME}$**, **$^BTS2_{M12E}$**, and **$^BTS2_{M21E}$**, respectively. These results suggest that the larger geometrical deformations led to the higher energies ($\Delta E_{TS}$ = 22.7 and 24.1 kcal/mol), further destabilizing the **$^BTS2_{M12E}$** and **$^BTS2_{M21E}$**. To confirm these steric effects, a topographic steric map analysis of catalysts **A** and **B** was carried out (Figure 10). As expected, the percent of buried volume of the metal center in catalyst **B** (% $V_{Bur}$ = 73.9) was significantly larger than that in **A** (% $V_{Bur}$ = 68.3), which is in line with the results of the

distortion–interaction analyses. The results obtained indicate that the catalytic performance was perhaps improved through modifying the catalyst with less sterically bulky ligands.

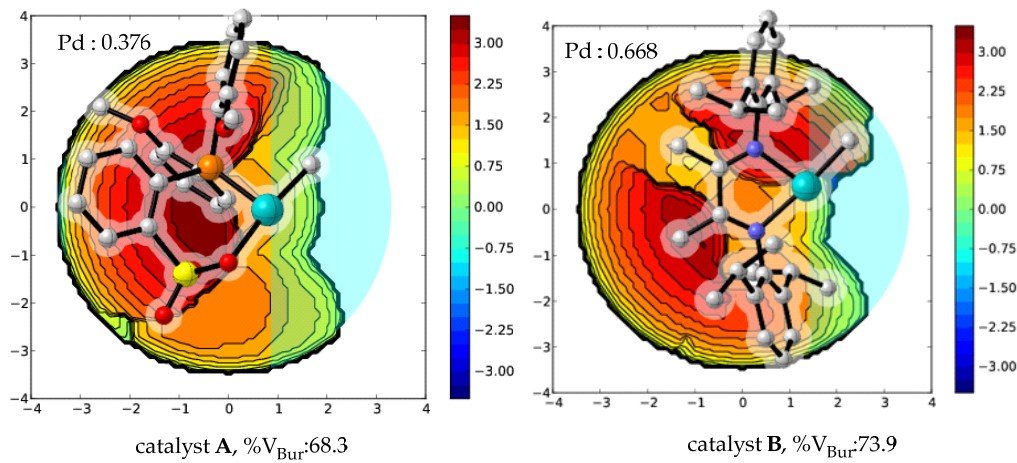

**Figure 10.** Topographical steric maps of catalysts **A** and **B**.

### 2.5. Catalyst Design

Inspired by the above discussions, in order to decrease the steric hindrance of catalyst **B,** the isopropyl in the phenyl moieties were replaced by a hydrogen atom (catalyst **B1**), a methyl group (catalyst **B2**), and an ethyl group (catalyst **B3**) (Figure 11), respectively, to improve the copolymerization activity of ethylene and MVS. Based on the more both kinetically and thermodynamically favorable $^{B}3_{M12}$, the chain propagation pathways via ethylene coordination ($^{B}3_{M12} \rightarrow {^{B}}4_{M12E}$) and insertion ($^{B}4_{M12E} \rightarrow {^{B}}TS2_{M12E} \rightarrow {^{B}}5_{M12E}$) by the new designed catalysts (**B1**, **B2**, and **B3**) were considered, in order to evaluate the catalytic performance of the new designed catalysts (**B1**, **B2,** and **B3**) (Figure S5, in the SM). To better clarify the issues at hand, the energy of the five-membered Pd$^{II}$ palladacycle intermediate $^{B}3_{M12}$ was used as a reference point. As shown in Table 1, the energies of ethylene coordination of catalysts **B1**, **B2**, and **B3** were 5.3, 6.4, and 6.7 kcal/mol, respectively, which were all lower than that (8.0 kcal/mol) of the original catalyst **B**. In addition, it was confirmed that the energy barriers of chain propagation (catalyst **B1**: 26.9, **B2**: 26.5, **B3**: 26.9 vs. **B**: 30.0 kcal/mol) were apparently reduced using the catalysts with less steric hindrance. To obtain more insight into the steric factor, a large sterically complex **B4** with *tert*-butyl group was designed. As anticipated, the higher energy of ethylene coordination (9.8 vs. 8.0 kcal/mol) was obtained by the larger steric catalyst **B4**. As expected, the buried volumes (%V $_{Bur}$) of catalysts **B1**, **B2**, **B3,** and **B4** increased in the order of **B1** < **B2** < **B3** < **B4** (Figure S6, in the SM). Beyond the steric modification, three catalysts with electron-withdrawing **B5**, **B6**, and **B7** were also designed (Figure 11). The catalysts **B5**, **B6**, and **B7** drastically decreased the ethylene coordination energies (2.9, 3.1, and 3.4 vs. 8.0 kcal/mol) (Table 1). In contrast to this slight change in the relative ethylene insertion energy barriers, the corresponding coordination energy drastically decreased from 8.0 to 2.9 kcal/mol in the new designed catalyst **B5**, which consequently reduced the chain propagation energy barrier. In an attempt to clarify the variation, the frontier orbitals of catalyst **B**, new designed catalysts **B1–B7**, and ethylene were analyzed. During the reaction, the LUMO of the electrophilic Pd catalyst interacted with the HOMO of ethylene.

As shown in Table 1, The HOMO energy (−7.60 kcal/mol) of ethylene was closer to the LUMO energies of species **B1–B3** and **B5–B7** (catalyst **B1**: −4.46, **B2**: −4.43, **B3**: −4.43, **B5**: −4.98, **B6**: −5.09, and **B7**: −5.11 kcal/mol), suggesting that ethylene coordination with the species **B1–B3** and **B5–B7** was easier than that with species **B** (−4.41 kcal/mol). Furthermore, a correlation coefficient ($R^2$) of 0.77 between the LUMO energies of catalysts **B** or **B1–B7** and the ethylene coordination energies was obtained by performing calculations (Figure 12). In summary, we proposed that the catalyst equipped with the lower

steric hindrance and electron-withdrawing group shows a lower energy barrier of chain propagation.

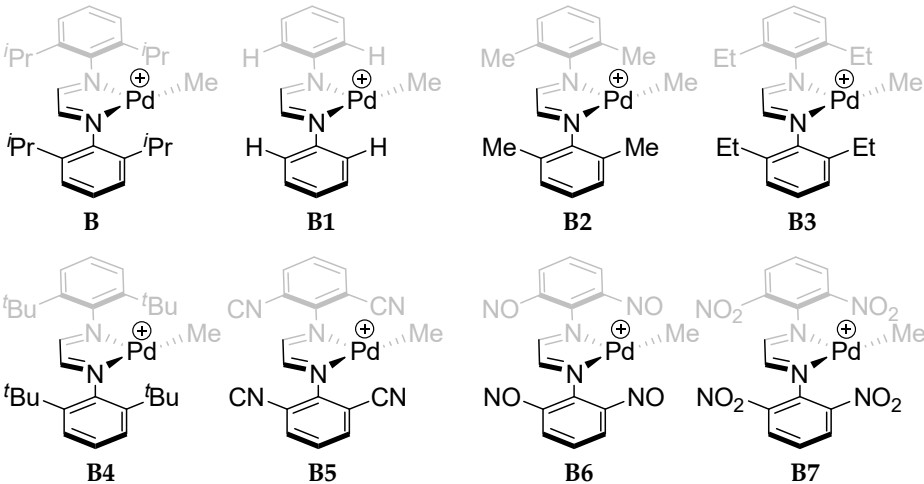

**Figure 11.** The initial species **B** and new designed catalysts **B1–B7**.

**Table 1.** Gibbs free energies (kcal/mol, relative to $^{B}3_{M12}$) for reactions of ethylene coordination and insertion, and frontier molecular orbital energies (kcal/mol) for catalysts **B** and **B1–B7** at the chain propagation stage.

| Catalysts | Coordination Energy | Relative Insertion Energy | Chain Propagation Energy Barrier | LUMO |
|---|---|---|---|---|
| **B** | 8.0 | 22.0 | 30.0 | −4.41 |
| **B1** | 5.3 | 21.6 | 26.9 | −4.46 |
| **B2** | 6.4 | 20.1 | 26.5 | −4.43 |
| **B3** | 6.7 | 20.2 | 26.9 | −4.43 |
| **B4** | 9.8 | 19.7 | 29.5 | −4.32 |
| **B5** | 2.9 | 21.2 | 24.1 | −4.98 |
| **B6** | 3.1 | 22.6 | 25.7 | −5.09 |
| **B7** | 3.4 | 20.7 | 24.1 | −5.11 |

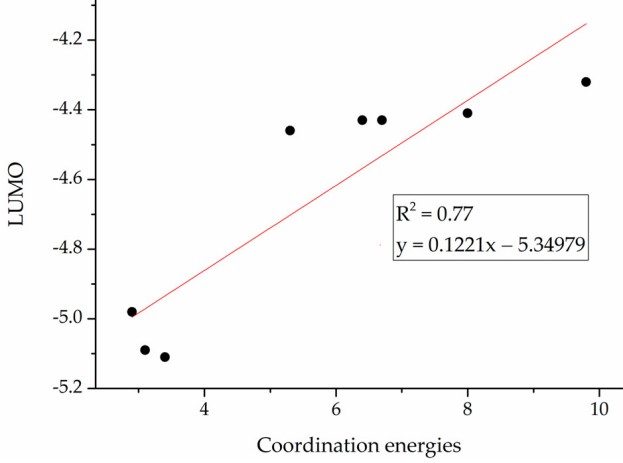

**Figure 12.** The $R^2$ values between the LUMO energies (kcal/mol) of catalysts **B** or **B1–B7** and their corresponding ethylene coordination energies (kcal/mol).

### 3. Computational Details

The calculations were performed using the G16 program. [32] The B3LYP-D3 [33]/BS1 method (BS1 = Lanl2DZ [34] for Pd and 6-31G(d,p) for all other atoms) was used for geometric optimizations and frequency calculations in the gas phase. In addition, a polarization function [35] for Pd [$\zeta$(f) = 1.472] was also added, and this provided qualitative, consistent results. For all of the transition states, intrinsic reaction coordinate (IRC) analysis [36] was performed to verify their identities. The B3LYP-D3/BS2 method (BS1 = SDD [37] for Pd and 6-311+G (d,p) for all other atoms) with the SMD model [38] (solvent = toluene) was employed for the solution-phase single-point energy calculations. We also tested the calculated solution-phase single point energies with other methods (M06-L-D3 [39], M06-D3 [40], B97D3 [40]) and used geometric optimization with the other basis set (SDD [37]) for some key steps in the chain initiation of two catalysts, and these methods showed similar performance (Figure S7 in the SM). These results were in line with those of the experimental studies. To estimate the importance of the dispersion corrections, comparisons were carried out with or without the addition of the Grimme's D3 dispersion [33] in the geometric optimizations and solution-phase single-point energy calculations, as shown in Figure S7 (in the SM). The structures in manuscript were visualized with CYLView. [41] The website tool SambVca was used to generate the topographic steric maps [42]. A free energy of solvation calculation was added to the thermodynamic corrections for the Gibbs free energy as well as 1.9 kcal/mol [43,44] (accounting for the standard state change from 1 atm (1 mol of an ideal gas) to 1 mol/L (1 mol/L in toluene solution) at 298.15 K), in order to obtain the final solution-phase Gibbs free energies for the following discussion. The root mean square deviations (RMSD) of the atomic positions were calculated using the VMD software [45].

### 4. Conclusions

The copolymerization of ethylene and methyl vinyl sulfone (MVS) catalyzed by phosphine–sulfonate and $\alpha$-diimine-based palladium catalysts were computationally investigated. In the case of the phosphine–sulfonate palladium catalyst, the 2,1-insertion manner of MVS was favored, which is attributed to a slightly twisted dihedral angle in the Pd-$C_1$-$C_2$-$C_3$ plane, and the lower geometric deformation. The phosphine benzene sulfonate palladium catalyst could produce a copolymer that vinyl polar monomers both incorporated into their main chains and chain ends because the insertion barrier of MVS was slightly lower than that of ethylene. It was found that the $\alpha$-diimine palladium catalyst **B** provided the most stable MVS insertion products, which could be ascribed to stronger O-backbiting interactions in the five- or four-membered $Pd^{II}$ palladacycle intermediate. Chain propagation has a rather high barrier; thus, our insight is that ethylene coordination would require a high energy cost to break the stable five- or four-membered chelate intermediates. Based on the previous discussion, $\alpha$-diimine palladium catalysts with a lower steric hindrance and an electron-withdrawing group were theoretically designed. The catalytic performance of the $\alpha$-diimine palladium catalysts was improved by the introduction of a group with less steric hindrance and an electron-withdrawing effect, reducing the LUMO energy of the catalyst molecule.

**Supplementary Materials:** The following supporting information can be downloaded at: https://www.mdpi.com/article/10.3390/catal13061026/s1, Figure S1: Calculated Gibbs free energy profile of *cis*/*trans* isomerization of ethylene mediated by **A** (distances in Å and energy in kcal/mol); Figure S2: Calculated Gibbs free energy profile of *cis*/*trans* isomerization of MVS mediated by **A** (distances in Å and energy in kcal/mol); Figure S3: Four possible transition states and corresponding intermediate free energies (in kcal/mol) for insertion of MVS into the Pd–Me bond by complex **A**; Figure S4: Overlay of catalyst and monomer in transition states and stable intermediates (a: 2,1-manner b: 1,2-manner) (in Å).; Figure S5: The considered chain propagation process based on MVS-inserted product ***B*3**$_{M12}$ by catalysts **B1**–**B7**; Figure S6: Topographical steric maps of catalysts **B** and **B1**–**B7**. The NBO charges on metal atoms are shown in black; Figure S7: Calculated relative Gibbs free energies by different basis sets and methods (in kcal/mol). The relative energies of the corresponding catalysts and monomers were set to be 0.0 kcal/mol; Table S1: Calculated thermodynamic corrections

for Gibbs free energies ($\Delta G_{cor}$ in Hartrees), solution-phase single-point energies ($\Delta E_{sol}$ in Hartrees) and solution-phase Gibbs free energies ($\Delta G_{sol}$ in Hartrees).

**Author Contributions:** Investigation, writing—original draft, writing—review and editing, L.Z. and X.K.; writing—review and editing, S.L. and W.Z.; conceptualization, supervision, writing—review and editing, Y.L. All authors have read and agreed to the published version of the manuscript.

**Funding:** This research was funded by the National Natural Science Foundation of China (No. 22071015, No. 22171038). X.K. thanks the Scientific Research Foundation of the Educational Department of Liaoning Province (LJKZ0848).

**Data Availability Statement:** The authors confirm that the data supporting the findings of this study are available within the article and its supplementary materials.

**Conflicts of Interest:** The authors declare no conflict of interest.

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
