# Peer review of "A DFT Study of the Copolymerization of Methyl Vinyl Sulfone and Ethylene Catalyzed by Phosphine–Sulfonate and α-Diimine Palladium Complexes"

_catalysts, doi:10.3390/catal13061026_

Round 1

Reviewer 2 Report

There are many typos that should be corrected

Round 2

Reviewer 1 Report

With this revised version, Luo and coworkers present a clear picture of the reactivity of methyl vinyl sulfone and ethylene. The revised version addresses most of the concerns I raised in the previous review. There is, however, a minor problem with Figure 12 and the accompanying text.

The authors show that the coordination energy correlates well with the HOMO-LUMO gap (HLG), which is calculated as the difference between the HOMO of ethylene and the LUMO of the catalyst. In the literature, the HLG is calculated with the energies of the frontier molecular orbitals of the same molecule, and combining orbitals of different molecules is not meaningful because orbitals are not physical observables. Using the HLG in Figure 12 corresponds to using a 'shifted' LUMO energy, and the 'shifting factor' is the HOMO energy of ethylene. Instead of using the HLG, the authors should consider reporting the correlation between the LUMO energy of molecules B–B7 and the respective coordination energy.

Reviewer 2 Report

The manuscript is improved, but the authors' comment regarding  Figure 2 can not be accepted, TS is never below the reactants, if this is the computational result, they have to write that the reaction is barrierless, exothermic reactions can be barrierless.
